# Assessing Time of Eating in Commensality Research

**DOI:** 10.3390/ijerph18062941

**Published:** 2021-03-13

**Authors:** Henrik Scander, Maria Lennernäs Wiklund, Agneta Yngve

**Affiliations:** 1School of Hospitality, Culinary Arts and Meal Science, Örebro University, 712 02 Grythyttan, Sweden; 2Department of Occupational and Public Health Sciences, Gävle University, 801 76 Gävle, Sweden; maria.lennernas@hig.se; 3School of Health Sciences, Örebro University, 702 81 Örebro, Sweden; agneta.yngve@oru.se; 4Department of Nutrition, Dietetics and Food Studies, Uppsala University, 751 22 Uppsala, Sweden

**Keywords:** eating together, conviviality, gastronomy, meal, food studies, dining, eating practice

## Abstract

Commensal meals seem to be related to a better nutritional and metabolic health as well as an improved quality of life. The aim of this paper was to examine to what extent research was performed using the search term commensality related to assessment of timing of meals. A scoping review was performed, where 10 papers were identified as specifically addressing the assessment of timing of commensality of meals. Time use studies, questionnaires, and telephone- and person-to-person interviews were used for assessing meal times in relation to commensality. Four of the studies used a method of time use registration, and six papers used interviews or questionnaires. Common meals with family members were the most common, and dinners late at night were often preferred for commensal activities among the working population. In conclusion, the family meal seemed to be the most important commensal meal. It is clear from the collected papers and from previous systematic reviews that more studies of commensal meals in general and about timing aspects in particular and in relation to nutritional health are essential to provide a solid background of knowledge regarding the importance of timing in relation to commensal meals.

## 1. Introduction

### 1.1. Meal Patterns and Grazing

In human nutrition and sociology, human eating behavior is referred to as *eating patterns* or *meal patterns*. Douglas & Gross (1981) define a meal as “A structured event that is a social occasion which is organized according to rules prescribing time, place and a sequence of actions. If food is taken as a part of the structured event, then we have a meal” [1]. Grazing assumes food is eaten in an incidental manner with respect to time, place, and content, responding to immediate preferences and desires [2]. Within nutritional research, commensality is emphasized as important for supporting nutritional health and food enjoyment [3,4]. Increasingly, it is apparent that meal timing must also be considered to ensure optimal health benefits in response to dietary patterns [5]. Moreover, commensality research is also important for an increased understanding of recent changes in dietary habits, social change and stability in the everyday eating [6] and eating out [7].

Does a natural meal pattern exist? In their review article, the authors [8] draw the conclusion that a basic meal pattern over time and culture is three main meals plus two subordinate meals per day. The authors also conclude that regular meals during the day help maintain the biological schedule of a circadian rhythm. In a 24-h society, regularity in living habits is sometimes abandoned due to shift work, jet lag, and social jet lag, and individuals might be active at all hours of the day [8]. An underlying assumption is that a social rhythm in eating that supports and strengthens the biological rhythms in hunger vs. satiation, while satiety promotes health, wellbeing, and optimal energy intake. In addition, commensality might have a supportive role in establishing a daily rhythm in eating.

### 1.2. Family Meals and Other Social Eating

The relationship between family meals and various nutritional health outcomes is an important research topic in relation to the great importance of healthy eating [9] or in relation to quarantine issues during the Covid-19 pandemic as pointed out by the WHO [10]. Family meals offer a rich opportunity to expose children to healthy foods, where children can learn healthy eating routines and behaviors from an early age [11]. A meta-analysis of the frequency of family meals showed a positive association between frequent family meals and better nutritional health. This was the case in younger and older children, across countries and socioeconomic groups, and for meals taken with the whole family vs. one parent. Regarding these findings, more research is recommended to target the causal direction of the relationship between family meal frequency and nutritional health [11]. The review also stated that most studies had investigated either ‘family dinner’ or ‘family meals’ and some to a lesser extent explored ‘family breakfast’ or ‘family lunch’. The authors concluded that the association between meal frequency and nutritional health holds regardless of whether families eat breakfast, lunch or dinner together.

Another review shows the inconsistency in the way in which studies have defined and assessed family meals [12]. The finding emphasized that family meals were associated with a range of positive outcomes among children and adolescents. Most studies used frequency or mean number of family meals per week. Less focus was directed towards different structural features that can also be of importance for family meals, such as who is present, eating place or setting, and how long family meals typically last. These findings provided an initial understanding of the structural features used to define family meals, but the findings also pointed to the importance of developing more comprehensive and sensitive assessment methods that can explain the complexity and multidimensional nature of family meals [12].

Poor health among the elderly and cognitively impaired is often related to a poor quality of life and problems with malnutrition and dehydration, as identified in a review [13]. The structuring of healthy eating and drinking was considered important in supporting healthy life patterns. The review found that studies were very limited, but the authors concluded that eating meals with caregivers, family meals or family style meals, soothing mealtime music, constantly accessible snacks, and longer mealtimes seemed to be promising interventions.

A recent systematic review of the relationship between nutritional behaviors and metabolic indices [14] identified “eating together” as one of the behavioral factors of importance. However, the review concluded that there were too few studies for a good analysis of the importance of eating together and recommended well-structured studies to disentangle links between nutritional behaviors and metabolic indices. Policymaking and the design of social interventions must rely on additional studies.

### 1.3. Different Patterns in Activity and Sleep

The circadian timing system or time-keeping system tells us when to wake up, eat, and go to bed. The “master clock” is the suprachiasmatic nucleus or nuclei (SCN) located directly above the optic chiasm in hypothalamus in the brain. SCN generates neural and hormonal rhythms that affect behavior as well as physiological functions [15]. Human beings, except for newborns, are “monocyclic”, which means that the 24-h day includes only two phases: activity and sleep. Exceptions are newborns, having more than one period of activity and sleep. A diminished timing system in the elderly is associated with decreased appetite and loss of thirst due to atrophy in the hypothalamic area. Joint cultural activities such as preparing food and commensality are difficult to maintain when some people work while others are asleep.

Humans may have a preference of waking up late and staying awake at night or waking up and going to bed early. This pattern might refer to their chronotype. A chronotype is the behavioral manifestation of underlying circadian rhythms in physiological processes. The two extremes are eveningness (delayed sleep period, “night owls”) and morningness (advanced sleep period, “larks”) [16]. Most individuals have some flexibility in the timing of their sleep period. Pre-pubescent children and the elderly often prefer waking up early (morningness), while adolescents prefer a delayed sleep period (eveningness). The appearance of different chronotypes within a family is demonstrated when infants, adolescents, parents, and grandparents prefer eating at different times of the day. Timing of family meals requires compromises.

### 1.4. Previous Studies of Shift and Night Workers’ Commensal Eating

Natural shifts between daylight and darkness naturally control the human rhythm in activity, sleep, and eating [15]. Travelling across time zones causes jet-lag due to the body’s circadian rhythm not being synchronized with the actual local day and night time. Dietary habits are considered a leading behavioral risk factor for human health in general, and there is growing evidence pointing to diet and sleep being related. For example, consumption of healthy foods has been shown to be associated with better sleep quality, while higher intake of processed and sugar rich foods was associated with worse sleep features [17]. The body cannot change the circadian rhythm immediately after moving between time zones for example, the adjustment takes place within a few days. Light, darkness, and social rhythm counteract the adjustment of the circadian rhythm in shift workers, leading to increased risk of a disruption, which is associated with impaired performance and metabolic disorders [18]. Shift workers often suffer from social jet-lag as a result of moving between two time zones: one that is governed by work and social obligations and another that is governed by our biological clock, also called the circadian time [19]. Among the workforce in Europe, 17 percent work shifts and 19 percent have night work [20]. In many studies, shift workers are defined as those who never work regular daytime hours [21]. During the last five decades, the proportion of shift workers has increased [22,23]. 

In a review, Lowden, A., et al. [23] examined 20 studies on food and nutrition among shift workers. The overview included dietary assessment studies. The results indicated that total energy intake over 24-h did not vary between day and shift workers or between work shifts. The paper presents guidelines for meal timing for shift workers and also general guidelines “provide appropriate dining facilities (…) allow a meal to be eaten away from the workplace, with colleagues, in as pleasant surrounding as possible”.

Nicholls et al. (2016) [24] investigated barriers and facilitators to healthy eating for nurses in the work place. Meals were often shared, and conversations about diet and exercise strengthened the motivation to adopt healthier habits. On the other hand, nurses often influenced each other to eat junk food, and social eating practices could involve “treats” such as cakes and pizza.

In a study using quantitative data [25], a longitudinal approach was used to explore the relative role of food-related, personal, and situational factors. The study monitored how 71 participants compiled and experienced 519 meals from their work place buffet during a three-month period. Meal satisfaction was directly associated with a positive ambience. Time available, mindful eating, and eating with close colleagues were positively associated with the perceived ambience. Among the most important factors was the possibility to share the meal with close colleagues.

In a qualitative semi-structured interview study [26], nine Scandinavian flight attendants were interviewed. The working conditions for this professional group often involves extended and irregular working hours, short rest periods, difficulties in planning for breaks, and high demands of service provision. Work schedules can include early shifts and time zone transitions that imply constant exposure to related health risks. Eating took place when food was available and when there was enough time to eat, rather than being guided by hunger and social context. Standing up to eat had almost become a marker of the fact that eating was not perceived as a part of a break. It became a part of a collegial agreement and a collegial act to wait until everyone had completed their working tasks. Eating together was often not possible.

In connection to performing an earlier scoping review on commensality [27], “time of eating” seemed to be a neglected topic, which is why an expanded search was performed on this topic. Scoping reviews are useful tools in the ever-increasing arsenal of evidence synthesis approaches, and one can conduct scoping reviews instead of systematic reviews where the purpose of the review is to identify knowledge gaps, scope a body of literature, clarify concepts or investigate research conduct (Munn et al., 2018). Therefore, our intent was to perform this scoping review as a precursor to other systematic reviews. This scoping review could serve as an important way to confirm the relevance of the topic and further develop the search methodology.

The aim of this paper was to examine study designs and methods used to assess timing of eating and commensality in studies of eating habits through a scoping review.

## 2. Materials and Methods

We performed a scoping review of papers discussing commensality assessment as a scoping review, mapping the body of a specific topic to summarize and disseminate relevant literature [28]. This scoping review followed the steps suggested by Arksey and O’Malley [29]: (1) identifying the research question, (2) identifying relevant literature, (3) selecting the studies, (4) charting the data, and (5) collating, summarizing and reporting the results. We used the search term commensality and searched the Web of Science Core Collection. Inclusion criteria were:Published up to 19 February 2021.Published in English.Somehow assessing commensality.Describing commensality among humans.Including aspects of timing.

A total of 168 papers were found in a search using the search term commensality. Ten papers were identified that especially discussed time in relation to commensal eating.

## 3. Results

In Table 1 below, the papers are presented in alphabetical order, while in the text we have described them depending on the method used.

### 3.1. Time Use Data in Commensality Research

Three of the studies were performed in Belgium by Mestdag et al. [30,31,32]. These studies used a method of time-use registration to collect information about commensality. A main feature was the family meal, but other commensal patterns were also researched. Two of these papers point out a substantial decrease of the number of family meals from 1966 to 1999, and the temporal boundaries of when to eat became more vague over time [30,31]. The authors state that the existence of a structured event in time and place was strongly impacted by the living arrangements and the move towards smaller households. Furthermore, cohabiting children were still having meals with their parents in 1999. A third study [32] questioned to what extent the role of the family meal was a reference point for shaping our eating habits. In a comparison of two Flemish time-use studies, from 1988 to 1999, the authors concluded that eating habits are highly structured events and are characterized as a social occasion. A Korean study, using a time-use survey pointed to an increase in social meals, often seemingly stretched out over time and including others than the close family [33].

### 3.2. Studies Using Questionnaires or Interviews on Commensality Research

Commensality in Japan was studied using face-to-face interviews [34]. This study showed that family commensality varied by co-habitation and time schedules for participants. The authors found no association between family meal frequency and household size. The study also showed that retired couples were able to eat common meals more frequently, while full-time employment made it more difficult to have commensal dinners on weekdays. The Japanese goal of having a family breakfast or dinner more than 11 times per week was demonstrated to be difficult to reach, especially among working people and non-nuclear families.

Three studies involved comparisons of routines over two cultures, one regarding French–British couples [35], the other two the cities of Santiago, Chile, and Paris, France [36,37]. The interview study on French and British couples revealed that the timing of dinners appeared to move against the temporal preference of the country of residence. The French in Britain particularly missed the lunch commensality that takes place much more between colleagues in France. The two Santiago–Paris studies performed through semi-structured questionnaires both showed that synchronicity of meals was positively influenced by the presence of others outside the close family circle and that dinner time needed some synchronization effort to create commensality, even within the family.

A British study was performed as an online survey presented to a sample of a consumer panel [38]. Older and more affluent, well-educated consumers were over-represented. The study showed that eating with household members was the most common commensal event. The authors also stated that even those living with others often ate alone. Weekends and evenings provided proof of more sociability at meals, while work meals were preferably eaten with colleagues.

A study from a Danish research group used two cross-sectional surveys performed in 1997 and 2012 in four Nordic countries [39]. The authors used telephone-based interviews or an internet-based questionnaire plus a 24-h recall to collect data on dietary quality. These dietary data provide a more realistic actual intake rather than the usual intake often used in research. This study showed some differences in regards to meal frequency, commensality, and meal composition from 1997 to 2012, but the differences were not the same in all four countries. More solitary meals and quick meals could be traced in some of the countries. The home and the workplace were the most frequent places for eating. Using phones and other electronic devices while eating was frequent in 2012, as well as eating in front of the TV. Living alone made a big difference in the increased number of commensal events. Dinner was the meal most likely to be enjoyed in company and took more time, even though sometimes eaten in front of a TV. The authors recommended the use of mixed methods, including more qualitative data from interviews, observations or open question surveys in order to arrive at a more comprehensive understanding of commensal eating as a whole and its importance for healthy choices.


*Making dietary surveys, one can use those that cover the “usual intake” such as using more or less complicated food frequency questionnaires (FFQ) with questions such as “how often do you eat certain foods?” The “actual intake” can be studied by methods where the research persons register their eating in a diary, sometimes using a scale or camera to document intake. They can also be interviewed about what they ate and drank during the last 24 h, or complete a questionnaire on what they ate and drank yesterday, sometimes with indications on when, where. The latter methods showing actual intake are more reliable than the FFQ’s usual intake, which becomes more vague and relies on memory. [40,41].*


**Table 1 ijerph-18-02941-t001:** Summary of the reviewed studies (*n* = 10): Time and Space.

Authors (Year), Country	Objectives	Study Design and Participants	Method	Results/Discussion	Results in Relation to the Review
Darmon & Warde (2019) [35] UK “Habits and orders of everyday life: commensal adjustment in Anglo–French couples”.	This paper examines processes of habit reshuffling and change in different contexts of household formation, looking specifically at habits regarding eating and commensality.	An interview study of 14 couples, each with one English and one French partner, half of whom live in France, half in England.	Interviewed couples together about their current eating habits, with follow-up interviews.	Making arrangements to eat together is one of the most fundamental and general facets of setting up a new household in Western societies. The importance of commensality for couples and families is widely recognized. The French particularly missed meals at lunch time, typically eaten with colleagues in France, but the biggest problems of adjustment for the migrant involved the rituals and timing of the commensal meals.	Showing how habits are scripted in three different orders of everyday life, governing diets, meal times, and extra-domestic commensality. The temporal and sociability orders appear to be geared to much firmer principles, governing performances in a more unified and ‘sticky’ way, and scripting more ‘solid’ incorporated habits, for both countries, where the key principles are the centrality of meal times and the table, to the collective time organization of the country of residence and the commensal and ritualized events.
Giacoman et al. (2021) [36] Chile and France “Meal synchronization and commensality in Santiago and Paris”.	This article analyzes the association between meal synchronization and commensality using representative survey data in Santiago and Paris.	Empirical material is drawn from two comparable data sources: *Encuesta de Comensalidad en Adultos de la Región Metropolitana* (Santiago, Chile) and *Santé*, *Inégalités* et *Ruptures Sociales* (Paris, France).	Santiago data: the survey was a semi-structured questionnaire about opinions and declared practices on commensality and sociodemographic characteristics, with a self-administered diary detailing all eating events. For Paris, the cohort study was a three-level random sample questionnaire.	In both metropolises, sharing meals with others more frequently was positively associated with having meals in synchronized timeslots. Next, we found differences between Santiago and Paris. In Paris, commensality was associated with synchronization in all three shared timeslots, and in Santiago, in the midday and the evening slots.	Sharing meals was positively associated with having meals at synchronized times. Enacting the social norm of eating on a regular schedule is likely to be influenced by the presence of others. Sharing a meal with others needs synchronization and facilitates the enactment of social norms around eating times.
Holm, L et al. (2016) [39] Changes in the social context and conduct of eating in four Nordic countries between 1997 and 2012.	Analyzes changes in the social organization of eating in four Nordic countries 1997–2012	Four Nordic countries, Denmark, Finland, Norway and Sweden. 4808 individuals aged 15 and older, 2012 8248 individuals 15–80 years of age	Two cross-sectional surveys, including the 24-h recall method. While the 1997 survey was based on computer-assisted telephone interviews, the 2012 survey was based on an internet-based questionnaire.	Some differences were seen between 1997 and 2012, but these were not consistent between countries. No dramatic changes in the social organization of eating on the whole.	Possibly a slacking of eating etiquette has taken place. In addition, more solitary meals and eating quickly; meals could be traced over the years in some countries. Using the 24-h recall method when collecting the data was considered giving a truer picture of the events rather than a general question of “usual” habits. Recommended use of a mixed-methods methodology including more qualitative data in parallel with the 24-h recall.
Kim (2020) [33] South Korea “Solitary eating, an inferior alternative? An examination of time-use data in South Korea”.	This study had two objectives: (1) to investigate the changes in the frequency, duration, and timing of solitary, family, and social meals in South Korea and (2) to examine the effects of these meals on subjective well-being.	Data from the Korean Time Use Survey, conducted by the Korea National Statistical Office every five years since 1999, analyzed in regards to meals.	Two-day time-diary survey in which respondents were asked to record their activities in a time diary consisting of 144 ten-minute time slots. The nationally representative sample varied in size across the four surveys.	South Koreans spent 83.8 min eating 2.69 meals on a typical weekday in 2014. The most noticeable change regarded family meals, which rapidly declined in both frequency (from 54.6% in 1999 to 39.8% in 2014) and total duration (from 53.6% to 38.4%). Solitary and social meals, on the other hand, showed substantial increases between 2004 and 2014; the former was conspicuous in its frequency, while the latter in length.	The shared family meal has declined rapidly in both frequency and duration. The increase in social meals explains why South Koreans spend more time eating, despite the declining frequency of meals. One explanation is that social gatherings in South Korea have increasingly involved eating and that the respondents reported the eating part of such socializing as time spent eating.
Lhuissier et al. (2020) [37] Chile and France “Meal times and synchronization: A cross-metropolitan comparison between Santiago (Chile) and Paris (France)”.	Investigation of meal schedules and their social determinants in order to question to what extent mealtimes are still socially shared events and what social institutions still shape national eating schedules.	Two comparable data sources: *Encuesta de Comensalidad en Adultos de la Región Metropolitana* (Santiago, Chile) and *Santé*, *Inégalités* et *Ruptures Sociales* (Paris, France), used to investigate meal schedules.	Cohort study representative of the adult (≥18) population in Paris and a survey representative of the adult (≥18) population living in the Santiago.	Highlighted cross-metropolitan similarities and disparities, regarding meal times and synchronization. Both metropolises shared a similar and marked three-meal pattern. Three major peaks distributed throughout the day correspond to breakfast, lunch, and dinner. Lunch was more synchronized than dinner, for reasons pertaining to professional and school rhythms. Dinner, however, demonstrated an important coordination effort towards the synchronization of social time within the family.	Collective mealtimes can be partly understood with regard national specificities. If meal time is tied to the constraints of social and professional life, it also expresses many other dimensions of the meal, including commensality. A comparison also highlighted important disparities between the two metropolises regarding meal schedules and synchronization.
Mestdag & Vandeweyer (2005) [30] Belgium ”Where has family time gone? In search of joint family activities and the role of the family meal in 1966 and 1999”	To investigate what place and time the family meal has within family time.	Time-use data used to study meal timing. Study 1965: 2077 Belgians 19–65 years; Study 1998–2000: 8392 Belgians 12–95 years old.	Registration of time-use data of daily practices. Two Belgian time-budget studies. Study I: one 24-h diary. Study II: One 24-h dairy during the working day and one 24-h diary for Saturday or Sunday. Diaries reported the total time spent on the most important family activities and meals as % of the total time of each activity. In addition, family meal timing was studied.	Time spent eating with partners and children were as follows in 1966 and 1999 (h, min): Workdays 0,51/0,27, Saturdays 1,05/1,03, Sundays 1, 23/1,05. Family time of day for meals peaked around 8 am, 1 pm, and 8 pm during working days, although fewer respondents were engaged in family meals 1999. Working days: family activities took place around eating times. Traditional meal times lost importance during 30 years. In 1999, meals were served later and fewer parents spent time with children at breakfast.	Family time spent was predominantly used for sharing meals. The number of commensal occasions with partners and children declined from 1966 to 1999; still, eating on every day of the week is the social par excellence for keeping up with the latest news and events of other family members. One underlying question is if family meals set the rhythm of family life.
Mestdag & Glorieux (2009) [31] Belgium ”Change and stability in commensality patterns: a comparative analysis of Belgian time-use data from 1996, 1999 and 2004”	To assess how commensality patterns have evolved in Belgium over the last decades and which factors have an impact on commensality.	The Belgian time-use survey 1966: 2076 Belgians, 19–65 years. 1999: The 2nd National Belgian time-use survey 8 382 Belgians, 12–95 years, from 4 275 households used to study commensal meals.	Time-use data of daily practices by use of a self-completion procedure using 10-min intervals. Two Belgian time-budget studies from 1966 and 1999. One 24-h diary (1966) and one diary during the working day and one 24-h diary for Saturday or Sunday (1999) Time data (2004) were used to determine the factors that affect commensality.	Average time spent on eating and relative share of total eating time on weekdays, Saturdays or Sundays, according to social context were reported. There was a significant decrease in commensality during the period: eating became more individualized. In 1966, 73% of weekday eating time occurred with at least one household member present, as compared to 56% in 1999.	Living arrangements had the strongest impact on commensality patterns. People who lived alone generally do not have anyone with whom they shared their meals. Married and cohabitant couples ate together on a regular basis; parents still shared a majority of meal times with their co-resident children.
Mestdag, I. (2005) [32] “Disappearance of the traditional meal: Temporal, social and spatial destruction.”	To study the disappearance of the “traditional” meal by comparing Flemish time budget data for 1988 and 1999. Temporal, social, and spatial features of the meal were studied.	Flemish time budget data were used to study meal features. In 1988, 463 respondents 21–40 years, in 1999, 599 respondents.	In 1988, respondents kept a diary for three consecutive days. In 1999, respondents followed the same procedure for one week. All activities including timing, duration, and location were registered. Respondents completed a questionnaire.	Flemish eating practices showed a high level of structure in the temporal, spatial, and organization of the meal. A clear three-meal pattern: breakfast, lunch, and dinner was observed both 1988 and 1999. The temporal boundaries of eating became vague, people ate less frequently at traditional meal times, the number or real meals decreased, and the number of informal snacks increased; indicators of temporal destruction were found.	The meals became shorter and solitary eating increased. Eating at home decreased in importance. In the introduction to the study, the following was discussed; a meal is when food is taken as a part of a structured event prescribing the time, place, and sequence of actions. Grazing assumes that food is taken in an incidental manner with respect to time, place, and content. It is unclear if the family meal still operates as a reference point in shaping eating habits.
Takeda, W., et al. (2018). [34] “Who eats with family and how often? Household members and work styles influence frequency of family meals in urban Japan.”	To examine frequencies of family commensality meals and the socio-cultural organization of eating and family lives.	Japanese adults aged 20–85 years in two metropolitan areas between 2009 and 2013 were interviewed regarding meals, including times. Lay people N = 179 and dietitians (N = 63).	242 surveys administered in face-to-face interviews in Tokyo and Kyoto. Interviews including open-ended free-list questions followed the mostly quantitative survey questions. Reports about usual meal times on working days and weekend days.	Peak weekday meal times: 7.00–7.59; 12.00–12.59; 19.00–19.59. Frequencies of family commensality are influenced by co-residents and work styles of participants rather than household sizes. Meal frequencies for family commensality were highest among those er 60 for all meals.	Work and lifestyle constraints impacting schedules appear to influence the frequency of family commensality. The Japanese government has promoted family commensality and set a goal; family breakfast and dinner more than 11 times per week. There were substantive gaps between the promoted image of family commensality and the practical and structural complaints on achieving family commensality, especially among working people and non-nuclear families. Full time workers had the latest average dinner time.
Yates L, Warde A. (2017) [38] Eating together and eating alone: meal arrangements in British households.	Examined meal arrangements in British households in 2012, drawing on an online survey in the format of a food diary administered to 2784 members of a supermarket consumer panel.	Investigating aspects of British meal patterns, provisioning and preparation, timing, and commensality. Online survey on meal arrangements	A small sample drawn from a consumer panel associated with a supermarket loyalty card scheme. The sample was 2784 individuals, i.e., a 45% response rate. Older, more affluent, better educated respondents and respondents without children were overrepresented.	Household members were the most common source of companionship in meals (75%) vs. work colleagues (16%).	Meals taking place later in the day were more likely to be eaten in companionship. Foods eaten with others are, for example, roasts, curry, fry-ups. When singles ate alone they were less likely to have substantial dishes than those who live with others but were eating alone. Adult-only households were underrepresented in this sample.

## 4. Discussion

Time-use studies, questionnaires, telephone- and person-to-person interviews were used for assessing meal times in relation to commensality. Commensality on weekday evenings was often hindered by fulltime employment, and commensal meals were often late in the evening. Commensal meals were often the result of planning and negotiation and sometimes even compromise regarding what to eat and when to eat.

### 4.1. Time-Use Studies

We found time-use studies from Belgium and Korea that took into account meal times and commensality patterns [30,31,32,33]. Time-use studies are often used in interdisciplinary research trying to identify how people allocate their time. Two National time-use surveys from the UK and the US, did not discuss commensality in their surveys [42,43], even though the US survey looked at time spent eating and in a particular wellbeing module asked about the feelings during that eating time. A National Swedish time-use survey incorporated lunch in the working hours, and the time after work was not specified for different chores or activities [44]. The time-use study method provides a clear picture of what happens during the day and can also outline sleeping habits in relation to time for breakfast, which then accentuates the possibilities for, for example, a family breakfast.

### 4.2. Studies Using Questionnaires or Interviews

Two studies compared results from commensality in different countries, with surveys in Santiago, Chile, and Paris, France [36,37]. Another study interviewed couples originating from France and the UK [35]. These types of cross-cultural studies are of immense importance in the current international community and can also be used to further understand differences between nationalities and ethnic groups in regards to meal times and related to commensality. Of course, etiquette in relation to timing of meals is of importance when hosting multicultural events or political dinners or meals in relation to business meetings, and a lot of effort is often put into these events [45]. This type of cross-cultural study provides a glimpse of important results that can be used for furthering our knowledge on commensality and meal timing etiquette in different cultures.

The Japanese study [34] based on face-to-face interviews showed that the national goal set for family meals at 11 per week, was almost impossible to reach. The fact that the Japanese government has promoted family commensality to such an extent that they have set a national goal for family breakfast and dinner is remarkable and would warrant further research for its applicability and usefulness in order to use it in other nations’ dietary guidelines.

The two remaining studies identified in this scoping review were very much alike [38,39], looking at adult eating habits, comparing two studies over time, and including an assessment of dietary quality. Both studies suffered from non-response problems, providing a sample which was skewed towards higher educated or in general, socio-economically more privileged groups. The British study [38] showed that commensal meals were about four times as common with household members compared to eating with colleagues. The study of the Nordic countries [39] showed that dinners were the most common meal eaten together. Over time, in some Nordic countries, there was a tendency for more solitary meals and more quick meals. Living alone made a big difference to the number of commensal meals [39], which has been shown in several previous studies [46]. It is important to note that largescale dietary surveys seldom take the opportunity to collect data on meal time or on commensality [43,44,47]. In previous studies from the workplace setting, eating together was mostly seen as a healthy eating behavior [24]. It was seen as important to negotiate meal break times with other colleagues, it was seen as a part of a collegial agreement and collegial act to wait until everyone had completed their working tasks [26]. Time available, mindful eating, and eating with colleagues were positively associated with commensal meals and were improved by a positively perceived ambience [25,36,37].

### 4.3. Aspects of Circadian Rhythm, Shift/Night Work, and When to Eat

Aspects of circadian rhythm or shift/night work and their relation to commensal eating were not mentioned in the ten papers identified by the search word commensality. This lack of research on the combined search terms is surprising especially due to the current interest in sleep as a main determinant of health [17]. The chronotypes of individuals in the family or extended family or factors related to shift and night work can be of importance for the negotiation of common meals. The time point for eating as well as for commensality are thus seldom covered in time-use studies or nutritional surveys. Guidelines on the time point for eating can be used to prevent obesity and metabolic disease [18,19]. 

Commensality at the workplace has a potential for increasing connection between work colleagues and to create an increased satisfaction with the work environment [25]. In studies of workplace meals during normal working hours as well as in shift work, several issues should be considered, including the time point, venue, and content of the meal, as well as the social context [23]. This type of research can provide a good background for health-promoting strategies in work place settings [24]. Commensality during shift work can have an important function when it comes to reducing tiredness during late night hours. Commensality and guidelines on the time point for food breaks and regular meal patterns can be strengthened by an increased knowledge of the biological rhythm of the human metabolism during odd hours. 

### 4.4. Strengths and Weaknesses of This Study and Suggestions for Future Research

This study used the search term commensality, which means that a number of studies that used other terms were not included. We could also conclude that the term commensality was not commonly used among researchers specifically studying timing for common meals. The strength of the study was that we used all papers found in the Web of Science Core Collection, which is a high-quality database that includes peer-reviewed papers from several different disciplines. We also did not limit the time for publication other than that the papers had to be published by February 2021. This paper is included in a project collection using a multi-disciplinary view of commensality [48].

In order to come up with scientifically trustworthy results, many more studies of good quality are needed to build a solid base for intervention design as well as for use when formulating policies and guidelines on commensal meal planning. Suggestions for future research could be, for example, the inclusion of timing as an aspect of commensal eating, performing many more high-quality studies, and preferably using a mixed-method including aspects of dietary quality as well as sociological and anthropological components. Studies of compromises for timing of meals could provide important information for guidelines, for example when it comes to late dinners due to work hours or other constraints or when studying commensal breakfasts over generational barriers with different chronotypes. Using the word commensality as a key word would ensure inclusion in studies of commensality. Since the studies identified in this scoping review came from Europe, South America and Japan, it seems as if the term has not been widely used in other parts of the world, at least not in studies also focusing on the time point of commensality. Aspects of circadian rhythm, shift work, and night work should be included in studies of common meals, including negotiating techniques for increasing the possibility of attendance of all generations or professionals. The inclusion of commensality and timing of meals in already ongoing large time-use studies and nutrition surveys would be easy and would provide valuable information. The use of good sampling methods is important to obtain a good view of sociodemographic factors, since many previous studies have a tendency of being skewed towards higher education, affluent, and older individuals.

## 5. Conclusions

Time-use studies in general as well as dietary surveys could easily introduce simple questions on commensality. The family meal seems to be the most important commensal meal, depending to a great extent on living arrangements. It is clear from the collected papers and from previous systematic reviews that more studies in the area of timing for commensal meals in relation to health are needed. In a number of studies looking at timing of meals collected in the introduction of this paper, the term commensality was not used. It would be of importance to perform a more comprehensive study using several search terms for commensal eating.

## Data Availability

Not Applicable.

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
