# Peer review of "Assessing Time of Eating in Commensality Research"

_ijerph, 2021, doi:10.3390/ijerph18062941_

Round 1
Reviewer 1 Report
[General findings]
This paper sought to understand the research trend on eating time of commensality. The theme may help to promote research on eating together with others. On the other hand, there were some questions about the research rationale, objective, and methods of this study. The following is a summary of some of the points of concern for your reference.
[Major]
L3: The title seems vague.
L50–92: I couldn't tell what you wanted to discuss one particular age or all ages.
L82–85: I think there have been many recent studies on eating alone and health status in the elderly.
L86–92: Why are you only discussing only the relationship with metabolic indices?
L116-171: Why are you only discussing work status among the many factors of eating habits?
L171–175: I couldn’t understand the rationale of examining it.
L175: I couldn’t understand the need to conduct a scoping review.
L177–178: The readers of this journal need to know the detailed search methods and review process. Without those descriptions, it is impossible to interpret the results and discussion.
L180: Isn’t the latest publication date a little old for a scoping review?
[Minor]
L354–356: Is this section still in the example text?
Author Response
Reviewer 1
[General findings]
This paper sought to understand the research trend on eating time of commensality. The theme may help to promote research on eating together with others. On the other hand, there were some questions about the research rationale, objective, and methods of this study. The following is a summary of some of the points of concern for your reference.
[Major]
L3: The title seems vague.
Answer: Thank you we have now worked out another title. “Assessing time of eating in commensality research”.
L50–92: I couldn't tell what you wanted to discuss one particular age or all ages.
Answer: We are trying to identify papers discussing assessment of time as a part of general assessment of commensality. This means commensal meals, including family meals, school meals, workplace meals and other social meals.
L82–85: I think there have been many recent studies on eating alone and health status in the elderly.
Answer: Thank you, we do agree, we are, however, concentrating on studies that use timing of meals as a part of commensality studies involving assessment methods. We have referenced some previous systematic reviews of family meals and other social meals including a systematic review on elderly in the introduction.
L86–92: Why are you only discussing only the relationship with metabolic indices?
Answer: Well, thank you, we identified a couple of systematic reviews for the introduction as a background for the review as such. This specific relationship is not the focus of the paper, we wanted to do a scoping review in order to identify which papers especially identified commensality and defined their assessment methods in relation to timing of meals.
L116-171: Why are you only discussing work status among the many factors of eating habits?
Answer: We are discussing a number of factors related to socioeconomic status, among those living conditions, such as living alone and work status.
L171–175: I couldn’t understand the rationale of examining it.
Answer: Thank you, that may be the most important remark. We understand your comment. We do hope that we met the criteria for making a scoping review in line with Munn et al and Arksey o’Malley. (added ref 27 and 28), and as a part of our project collection using a multi-disciplinary view of commensality, described in the Editorial paper Yngve et al (added ref 47). A scoping review provides a quick and dirty overview of a research field, and you can use it when you want to prepare for a full systematic review, or in order to understand a concept.
Munn, Z., et al., Systematic review or scoping review? Guidance for authors when choosing between a systematic or scoping review approach. BMC medical research methodology, 2018. 18(1): p. 1-7.
Arksey, H. and L. O'Malley, Scoping studies: towards a methodological framework. International Journal of Social Research Methodology, 2005. 8(1): p. 19-32.
Yngve, A., et al., The Project Collection Food, Nutrition and Health, with a Focus on Eating Together. International Journal of Environmental Research and Public Health, 2021. 18(4): p. 1572.
L175: I couldn’t understand the need to conduct a scoping review.
Answer: Scoping reviews are useful tools in the ever increasing arsenal of evidence synthesis approaches and one can conduct scoping reviews instead of systematic reviews where the purpose of the review is to identify knowledge gaps, scope a body of literature, clarify concepts or to investigate research conduct…. scoping reviews can also be helpful precursors to systematic reviews and can be used to confirm the relevance of inclusion criteria and potential questions as pointed out by MUNN et al 2018 (added ref).
L177–178: The readers of this journal need to know the detailed search methods and review process. Without those descriptions, it is impossible to interpret the results and discussion.
Answer: Absolutely, we agree it was a bit too short, we have added more details on the scoping review process. Line 201-205
as a scoping review mapping the body of a specific topic to summarize and disseminate relevant literature [27] This scoping review followed the steps suggested by Arksey and O’Malley [28]: (1) identifying the research question, (2) identifying relevant literature, (3) selecting the studies, (4) charting the data and (5) collating, summarizing and reporting the results.
L180: Isn’t the latest publication date a little old for a scoping review?
Answer: We have now updated the search with papers up to yesterday’s date, the 19th of Feb 2021. This new search gave us four additional papers, which are now included in the scoping review.
[Minor]
L354–356: Is this section still in the example text?
Answer: Yes, thank you for noticing, that is correct, we have now taken that part out.
Reviewer 2 Report
Certainly, one of the strengths of this manuscript is its novelty, especially given recent changes in the dietary habits/diet quality and rise in the prevalence of non-communicable diseases.The are several issue that should be addressed by the authors before the final publication.
Recent changes in dietary habits could be better described: a transit from traditional dietary patterns to western like dietary patterns accompanied by a rise in the prevalence of chronic diseases and worsened sleep quality.
Authors should better describe link between diet and circadian rhythm and sleep in the paragraph “Different patterns in activity and sleep”, please see doi.org/10.1016/j.smrv.2021.101430.
A brief summary of the effects of timing of the meal toward human health would be welcome.
Table 1. There is no need to repeat names of the columns. Please include a column (or add to existing column) with study design.
It is not clear whether the study is a narrative review or systematic review. Please provide a clear statement. If authors aimed to perform systematic review further information on the methodology is needed.
Author Response
Reviewer 2
Certainly, one of the strengths of this manuscript is its novelty, especially given recent changes in the dietary habits/diet quality and rise in the prevalence of non-communicable diseases. The are several issue that should be addressed by the authors before the final publication.
Recent changes in dietary habits could be better described: a transit from traditional dietary patterns to western like dietary patterns accompanied by a rise in the prevalence of chronic diseases and worsened sleep quality.
Answer: We have added a few sentences on this in the introductory part: Line: 43-45
Moreover, commensality is also important for an increased understanding of recent changes in dietary habits through the studies of social change and stability in the everyday eating [6] and eating out [7].
Authors should better describe link between diet and circadian rhythm and sleep in the paragraph “Different patterns in activity and sleep”, please see doi.org/10.1016/j.smrv.2021.101430.
Answer: Absolutely, we agree, this link is better described now, we have reformulated a part of the introduction. Line 135-140. (added ref 17).
A brief summary of the effects of timing of the meal toward human health would be welcome.
Answer: Thank you we have now extended on this with a few sentences for the introduction as well. Line 39-42
Within nutritional research, commensality is emphasized as important for supporting nutritional health and food enjoyment [3, 4]. Increasingly it is apparent that meal timing must also be considered to ensure optimal health benefits in response to dietary pattern [5]
Table 1. There is no need to repeat names of the columns. Please include a column (or add to existing column) with study design.
Answer: Thank you for this comment, we have now included a column on study design, and we have removed the repeat names of the columns.
It is not clear whether the study is a narrative review or systematic review. Please provide a clear statement. If authors aimed to perform systematic review further information on the methodology is needed.
Answer: This is a scoping review following the steps suggested by Arksey and O’Malley (added ref 23 added), and in line with Munn et al. (added ref 22). Line 201-205.
We performed a scoping review of papers discussing commensality assessment, as a scoping review mapping the body of a specific topic to summarize and disseminate relevant literature [27] This scoping review followed the steps suggested by Arksey and O’Malley [28]: (1) identifying the research question, (2) identifying relevant literature, (3) selecting the studies, (4) charting the data and (5) collating, summarizing and reporting the results.
Reviewer 3 Report
The manuscript entitled "Eating time in commensality research" is a scoping review that looks at the research done in commensality with respect to the timing of meals. The subject will be of great interest to the general public and important for establishing policies as stated by the authors. However, below are the major comments regarding the review.
Major comments:
1) The review covers only 6 major studies and is based on the search for one term "commensality" in a limited database "Web of Science". Why were other terms that are related such as "family meal" not included and other databases not looked at? The need for more studies is evident in the repetition of sentences regarding the studies. I have stated one example below and there are others
Example:
Lines 220-222- "The study showed that eating with household members was most common, almost four times as common as eating with work colleagues."
and
Lines 283-285- "The British study [25] showed that commensal meals were about 4 times as common with household members compared to eating with colleagues."
2) There is a general vagueness about the language of the review. While it may be a scoping review, at times the authors give too many details in the results section that are already mentioned in the table, and other times as in the example below there are absolutely no details.
Lines 226-228-"The authors used a 24-hour recall to collect data also on dietary quality, based on a more realistic actual intake rather than the usual intake often used in research." Here what do the words "more realistic actual intake mean?
The review needs to be shorter, better structured, and more succinct.
Author Response
Reviewer 3
The manuscript entitled "Eating time in commensality research" is a scoping review that looks at the research done in commensality with respect to the timing of meals. The subject will be of great interest to the general public and important for establishing policies as stated by the authors. However, below are the major comments regarding the review.
Major comments:
1) The review covers only 6 major studies and is based on the search for one term "commensality" in a limited database "Web of Science". Why were other terms that are related such as "family meal" not included and other databases not looked at?
Answer: We spent some time on producing an introduction covering family meals and other social meals as well as workplace meal studies. The scoping review in itself was only able to show studies that used commensality as the search term that described their assessment methodology and that included timing of meals in their method. We used the scoping review as our method and this paper is a part of a project collection on diet, nutrition and health in relation to eating together. We could see that the definitions of “family meal”, “eating together” and “social eating” were not very well developed. We wanted to get a quick overview of the field when using only the search term commensality. The overlap with the different search terms is certainly important as you can see from the few papers we found. The database Web of Science Core Collection is one of the best there is, and the number of journals that are included from a multitude of research areas is impressive and increasing over time. We chose this database since we found it reliable when it comes to high quality peer-reviewed papers being included. In Sweden, this database is the one that our universities rely on when they get evaluated for funds from the government and when our departments get their local funding or apply for research grants. As a scoping review, it is supposed to provide us some guidance on the issue at hand. We can certainly see that there needs to be:
1) more studies performed on assessment of timing in commensality 2) more systematic reviews performed, which then highlights 3) a need for better definitions for all terms connected to commensality, such as family meals, eating together, social eating etc. and also requires 4) a large number of studies performed with a high quality. Quality of studies is not judged in scoping reviews.
The need for more studies is evident in the repetition of sentences regarding the studies. I have stated one example below and there are others
Example:
Lines 220-222- "The study showed that eating with household members was most common, almost four times as common as eating with work colleagues."
and
Lines 283-285- "The British study [25] showed that commensal meals were about 4 times as common with household members compared to eating with colleagues."
Answer: In regards to the duplication of statements, we have of course deleted one of them.
2) There is a general vagueness about the language of the review. While it may be a scoping review, at times the authors give too many details in the results section that are already mentioned in the table, and other times as in the example below there are absolutely no details.
Lines 226-228-"The authors used a 24-hour recall to collect data also on dietary quality, based on a more realistic actual intake rather than the usual intake often used in research." Here what do the words "more realistic actual intake mean?
Answer: Absolutely, thank you. When nutritionists make dietary surveys, they can use those that cover the “usual intake” such as “how often do you eat certain foods?” or you can use a method where the research persons register their eating in a diary, sometimes using a scale to weigh all food items or using a camera to document for example. They can also be interviewed about what they ate and drank during the last 24 hours, or complete a questionnaire on what they ate yesterday. The latter methods are more reliable than the former with usual intake, which becomes more vague and relying on memory. The usual intake does normally not take into account seasonal variations of foods for example. We have added a few lines describing the differences in the paper: Line 299-307.
“Making dietary surveys, one can use those that cover the “usual intake” such as using more or less complicated food frequency questionnaires (FFQ) with questions such as “how often do you eat certain foods?” The “actual intake” can be studied by methods where the research persons register their eating in a diary, sometimes using a scale or camera to document intake. They can also be interviewed about what they ate and drank during the last 24 hours, or complete a questionnaire on what they ate and drank yesterday, sometimes with indications on when, where. The latter methods showing actual intake are more reliable than the former with usual intake, which becomes more vague and relying on memory. [39, 40]. “
The review needs to be shorter, better structured, and more succinct.
Answer: Thank you for this comment. We have now updated the search to include four more papesr identified in an identical search on Web of Science up to the 20th of Feb 2021. We have also worked on the structure of the tables, we have cut a bit in the text and explained a bit more about the context for this paper in the project collection. We hope that you now find it a bit easier to read and more up to par with your suggestions.
Round 2
Reviewer 1 Report
The authors answered the rationale and necessity of the scoping review, but I could not read of the rationale and necessity of this study itself from the manuscript. I was also unable to confirm response on how to conduct a detailed scoping review (e.g., search strategy, process for selecting sources, whether data charting was done independently or in duplicate).
Author Response
Reviewer 1: The authors answered the rationale and necessity of the scoping review, but I could not read of the rationale and necessity of this study itself from the manuscript. I was also unable to confirm response on how to conduct a detailed scoping review (e.g., search strategy, process for selecting sources, whether data charting was done independently or in duplicate).
Response: Thank you for these comments, we have clarified the rationale, necessity and the methodology for scoping review as follows:
“In connection to performing an earlier scoping review on commensality (Scander et al. 2021), “time of eating” seemed to be a neglected topic, which is why an expanded search was performed on this topic. As scoping reviews are useful tools in the ever-increasing arsenal of evidence synthesis approaches and one can conduct scoping reviews instead of systematic reviews where the purpose of the review is to identify knowledge gaps, scope a body of literature, clarify concepts or to investigate research conduct (Munn et al., 2018). Therefore, our intent was to perform this scoping review as a precursor to other systematic reviews. This scoping review could therefore serve as an important way to confirm the relevance of the topic and further develop the search methodology.” Line 248-258
Reviewer 2 Report
I would like to thank the authors for providing a revised version of their manuscript.
Author Response
Reviewer 2: I would like to thank the authors for providing a revised version of the manuscript
Response: Thank you for reading and commenting on the first version! Very helpful.
Reviewer 3 Report
The manuscript is significantly improved and reads well.
Only correction Line 238- Check spelling of preference
Author Response
Reviewer 3: The manuscript is significantly improved and reads well. Only correction Line 238 –Check spelling of preference.
Response: Thank you for your kind words, we have corrected the spelling on line 238.